# MEMORIZING TRANSFORMERS

Yuhuai Wu, Markus N. Rabe, DeLesley Hutchins, Christian Szegedy

`{yuhuai,mrabe,delesley,szegedy}@google.com`

## ABSTRACT

Language models typically need to be trained or finetuned in order to acquire new knowledge, which involves updating their weights. We instead envision language models that can simply read and memorize new data at inference time, thus acquiring new knowledge immediately. In this work, we extend language models with the ability to memorize the internal representations of past inputs. We demonstrate that an approximate $k$NN lookup into a non-differentiable memory of recent (key, value) pairs improves language modeling across various benchmarks and tasks, including generic webtext (C4), math papers (arXiv), books (PG-19), code (Github), as well as formal theorems (Isabelle). We show that the performance steadily improves when we increase the size of memory up to 262K tokens. On benchmarks including code and mathematics, we find that the model is capable of making use of newly defined functions and theorems during test time.

## 1 INTRODUCTION

Transformers (Vaswani et al., 2017) have led to remarkable progress in natural language processing (Devlin et al., 2019; Brown et al., 2020), mathematical reasoning (Polu & Sutskever, 2020; Wang et al., 2020a; Rabe et al., 2021; Li et al., 2021; Hahn et al., 2021; Cobbe et al., 2021), and program synthesis (Austin et al., 2021; Chen et al., 2021; Li et al., 2022). However, transformer performance on many of these tasks is limited by the context length of attention, which is typically short. The ability to attend to far-away tokens is important in many situations. In novels, characters and events are referenced across multiple chapters. In source code, references to classes and functions may occur quite far from the places in which they are defined. In theorem proving, proofs make use of previously defined lemmas.

Attention over long sequences is also useful as a form of rapid learning. Facts and information which are stored in the form of weight matrices must be slowly trained over hundreds of thousands of training steps. By using attention, however, a model can simply *memorize* facts (e.g. function definitions) by storing them as (key, value) pairs in long-term memory, and then retrieve those facts later by creating a query that attends to them. In this case, attention acts as a form of information retrieval, allowing the model to look up facts that it has seen previously.

We demonstrate that a simple and effective way to increase the size of the attention context is to use approximate $k$-nearest-neighbor ($k$NN) lookup, which is widely used in information retrieval. A number of extremely scalable implementations of $k$NN lookup are available, such as ScaNN (Guo et al., 2020) and Faiss (Johnson et al., 2021).

There are two things which distinguish our approach from previous work on long-range attention (c.f. Section 2). First, unlike some other approaches, $k$NN lookup does not do averaging or summarization of tokens at long distances, but retrieves exact values even from the distant context.

Second, gradients are not backpropagated into the external memory, which is critical to the scalability of our technique. The keys and values are a function of model parameters, so attempting to backpropagate gradients into external memory would necessarily involve computing all of the keys and values with the current model parameters on every training step. However, if the external memory is not differentiable, then we can instead instead reuse keys and values that were previously computed on prior training steps, which drastically reduces the amount of computation for large memories. With

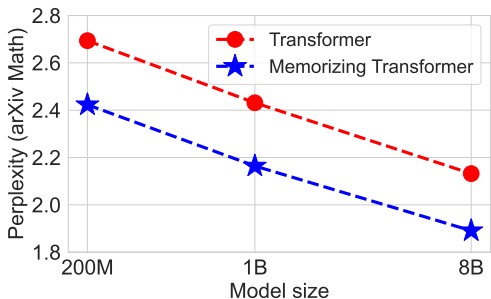

Figure 1: Adding a memory of 8K tokens improves perplexity across different model sizes.

our technique, we are easily able to scale external memory up to sequence lengths of 131k or 262k tokens on a single TPU device, while maintaining a reasonable step time.

We show that model perplexity steadily improves with the size of external memory on a variety of language modelling tasks, including C4 (long documents only), Github code repositories, PG-19 books, formal proofs in Isabelle, and arXiv math papers. We further show that models can generalize to larger memory sizes than they were trained on: models trained with a small memory show gains from using a much larger memory at inference time. Finally, we show that our models are actually using memory in the way that we had hoped, e.g. by looking up the definitions of lemmas in a theorem proving corpus.

The simplicity of the changes to the Transformer architecture allows us to easily integrate this approach into existing code bases, including extremely large language models. We further show that the improvements to quality are maintained across models of increasing size, and that the model improvements gained from adding memory are even larger than increasing the size of the model by 5X or more as shown in Figure 1.

## 2 RELATED WORK

A great deal of work has been done on efficient long-range attention mechanisms; see Tay et al. (2020; 2021) recent surveys. Sliding windows (Beltagy et al., 2020) use a long sequence, but attend within a smaller window, thus reducing complexity to the window size, rather than total sequence length. Approximate mechanisms such as Linformer (Wang et al., 2020b), and Performer (Choromanski et al., 2021) refactor the attention matrix by using a different kernel than softmax to obtain $O(N)$ complexity. Pooling strategies such as Hierarchical 1D attention (Zhu & Soricut, 2021), and Combiner (Ren et al., 2021) apply pooling or averaging over tokens at longer distances. Sparse strategies such as Big Bird (Zaheer et al., 2020) select only a subset of tokens to attend to; Routing Transformers (Roy et al., 2021) use clustering to select the subset, while Reformer (Kitaev et al., 2020) relies on hashing. Hierarchical mechanisms (Ainslie et al., 2020) combine multiple tokens into phrases or sentences to reduce sequence length. Expire-span (Sukhbaatar et al., 2021) prunes far-away tokens that it learns are "unimportant". (Zemlyanskiy et al., 2021) process long sequences in two passes with different encoders. The second pass is given a lot of context by accessing summaries of the first pass.

Feedback transformers (Fan et al., 2020) use a recurrent architecture in which each token attends to the output of the final layer instead of the previous layer. Recurrence does not increase the size of the attention context itself, but it expands the receptive field at the cost of parallelism and training speed.

Truncated backpropagation through time (Williams & Peng, 1990) was originally introduced as a way of training recurrent neural networks (RNN) over very long sequences, when the entire sequence does not fit in memory. The sequence is chopped into segments, and after each training step, the final RNN state for the segment is saved in a non-differentiable cache, and used as the initial state on the next training step. Neural caches (Grave et al., 2017) extend the cache to contain a record of many prior hidden states, and attend over them. Transformer-XL (Dai et al., 2019) applies this technique to transformers; it caches the (key,value) pairs computed from the previous training step, and uses them as a prefix for the tokens on the next training step, which yields significant gains on long documents. Rae et al. (2020) improve over Transformer-XL by compressing the tokens before adding them to the

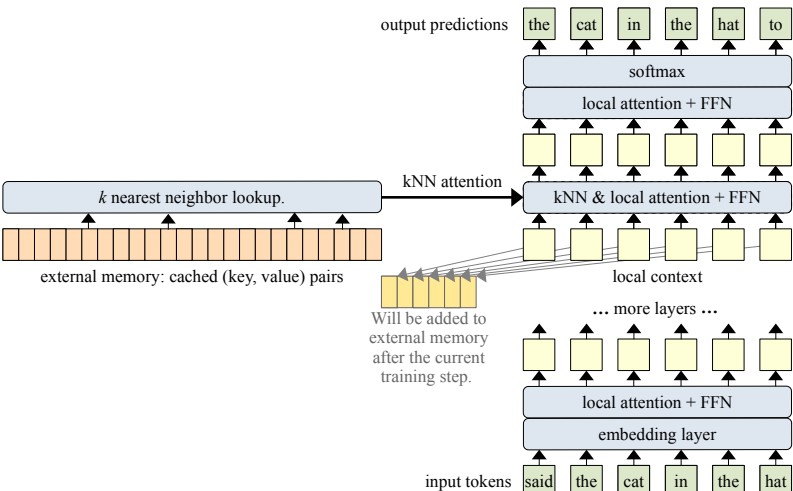

Figure 2: We extend Transformers with access to (key, value) pairs of previously seen subsequences.

cache. In contrast, we use a very large cache without compression, combined with an approximate $k$NN attention mechanism over it.

Sukhbaatar et al. (2019) make the observation that the feed-forward portion of a transformer layer functions very much like attention if one replaces the ReLU activation with softmax. They implement a combined attention over both tokens from the input sequence and a learned (and differentiable) "memory". Lample et al. (2019) exploit this observation to replace the feed-forward layers (FFNs) with a fast $k$NN lookup over a much larger "memory", and achieve large gains in model accuracy without significant computation overhead. (We use $k$NN lookup to approximate attention to previous tokens, not to replace the FFN.)

Non-differentiable external memory has been used in different ways by Khandelwal et al. (2020), who run a pre-trained model over an entire corpus, and construct a large table of (key, token) pairs. They then use that table to replace the final softmax layer for token selection in the model, which results in significant improvements in language modeling. Yogatama et al. (2021) extend this approach by a gating mechanism and a process to compress the context into keys for retrieval.

There are several works that combine retrieval with transformers. REALM (Guu et al., 2020), MARGE (Lewis et al., 2020a), RAG (Lewis et al., 2020b), and composite memory for dialog (Fan et al., 2021) retrieve documents from a knowledge base to improve question answering or dialogue. The knowledge base consists of text snippets and is static and typically separate from the inputs and outputs of the models. Instead, we focus on language modeling using a decoder-only model, and propose a simple model that *unifies attention and retrieval*.

$k$-nearest-neighbor lookup is a general-purpose technique that is used for a wide variety of machine learning and retrieval tasks, and high-performance implementations are available for various architectures (Johnson et al., 2021; Guo et al., 2020). Memory-efficient Transformers (Gupta et al., 2021) replace dense attention with a $k$NN lookup to increase speed and reduce memory usage.

## 3 METHOD

The architecture of our $k$NN-augmented transformer is shown in Figure 2. The bulk of the model is a vanilla, decoder-only transformer (Vaswani et al., 2017). The input text is tokenized, and the tokens are embedded into vector space. The embedding vectors are passed through a series of transformer layers, each of which does dense self-attention, followed by a feed-forward network (FFN). Since this is a decoder-only language model, we use a causal attention mask and the token embeddings of the last layer are used to predict the next token.

Long documents are split into subsequences of 512 tokens, and each subsequence is used as the input for one training step. In contrast to standard practice, we do not shuffle the subsequences; instead, each long document is fed into the transformer sequentially, from beginning to end, as is done with Transformer-XL (Dai et al., 2019).

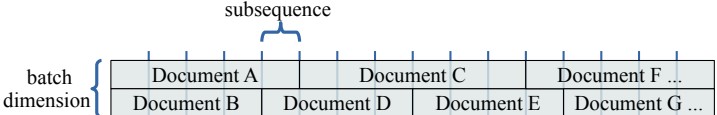

Figure 3: Our data pipeline splits documents into subsequences and packs subsequences into batches.

We also use a Transformer-XL style cache, which holds the keys and values from the previous training step. When doing self-attention, the cached keys and values are prepended to the current keys and values, and we use a sliding-window causal mask (Beltagy et al., 2020) so that each token has a local context that includes the previous 512 tokens.

## 3.1 $k$NN-AUGMENTED ATTENTION LAYER

One of the transformer layers near the top of the stack is a *kNN-augmented attention layer*, which combines two forms of attention. Like all of the other layers, it uses standard dense self-attention on the *local context*, which is the input subsequence for the current training step. Unlike the other layers, however, it also does an approximate $k$-nearest-neighbor search into the *external memory*.

The same queries are used for both the local context, and for the external memory. The keys and values also belong to the same distribution; after each training step, the (key, value) pairs in the local context are appended to the end of the external memory. If the document is very long, old (key, value) pairs will be dropped from the memory to make room for new ones. Thus, for each head, the external memory keeps a cache of the prior $M$ (key, value) pairs, where $M$ is the memory size.

The $k$NN lookup will return a set of *retrieved memories*, which consist of the top-$k$ (key, value) pairs that $k$NN search returns for each query (i.e. each token) in the input subsequence. As with standard dense attention, we first construct an attention matrix by computing the dot product of each query against the retrieved keys, then apply softmax, and finally return a weighted sum of the retrieved values. Unlike standard dense attention, the retrieved memories contain a different set of (key, value) pairs for each query.

Attention over the local context is performed in the usual way. The results of $k$NN-attention and local attention are then combined using a learned gate:

$$g = \sigma(b_g) \tag{1}$$
$$\boldsymbol{V}_a = \boldsymbol{V}_m \odot g + \boldsymbol{V}_c \odot (1 - g) \tag{2}$$

where $\sigma$ is the sigmoid function, and $\odot$ is element-wise multiplication. $\boldsymbol{V}_a$ is the combined result of attention, $\boldsymbol{V}_m$ is the result of attending to external memory, and $\boldsymbol{V}_c$ is the result of attending to the local context. The *bias* $b_g$ is a learned per-head scalar parameter, which allows each head to choose between local and long-range attention. In our experiments, the value of the gate $g$ does not depend on the content of the token at each position, although that would be a trivial extension to implement. We did observe that over time, most heads learned to attend almost exclusively to external memory.

**Position bias.** For dense attention within the local context, we use the T5 relative position bias (Raffel et al., 2020). As noted by Dai et al. (2019), adding a global position encoding to each token does not work well when processing long documents. We don't use a position bias for the retrieved memories. Experiments on the PG19 dataset (Sun et al., 2021) have shown that relative position does not appear to matter at long range, and the T5 relative bias puts all long-range tokens in the same bucket anyway.

**Batching.** Figure 3 illustrates how multiple long documents of different lengths are packed into a batch, and split into subsequences. Each subsequence in the batch comes from a different document, and thus requires a separate external memory, which is cleared at the start of each new document.

## 3.2 DISTRIBUTIONAL SHIFT

Because each long document is processed over multiple training steps, there is a distributional shift in the keys and values that are stored in external memory. The model parameters that produce the queries change over time, and will thus have shifted since the keys and values were stored. For very large memories, older records may become "stale." Similar observations have been made for CrossBatch memory (Wang et al., 2020c) in the vision domain.

To reduce the effects of staleness, we normalize keys and queries (Henry et al., 2020). Normalization does not eliminate staleness, but it at least ensures that older keys and newer keys do not differ in magnitude. We also found that normalization helps stabilize training with the Transformer-XL cache.

In some of our experiments, we observed that training models from scratch with a large memory sometimes resulted in worse performance than pretraining the model with a small memory of size 8192, and then finetuning it on a larger memory. This training instability could be due to staleness. However, models seem to be able to cope with a limited degree of staleness (with the small memory) by adjusting their queries accordingly.

### 3.3 Approximate $k$NN

We employ *approximate $k$NN* search rather than exact $k$NN search because it significantly improves the computational speed of our model. We use a simple approximation of $k$NN for TPUs, which has a recall of about 90%, i.e. 90% of the true top $k$ are returned in the approximate top $k$. There are various other efficient approximate $k$NN algorithms available for CPU and GPU/TPU, for example through Faiss (Johnson et al., 2021) or ScaNN (Guo et al., 2020), which can scale into the billions.

## 4 Experiments

We evaluate the effect of adding external memory on five language modeling tasks, all of which involve long-form text: English language books (PG-19), long web articles (C4), technical math papers (arXiv Math), source code (Github), and formal theorems (Isabelle). The results show significant improvements in the perplexity of the model with the addition of external memory. We experimented with various sizes of external memory, from 1536 to as high as 262K. On most of the datasets, there was an initial sharp gain from adding a small external memory, followed by smaller but steadily increasing gains as the size of the memory was increased.

### 4.1 Datasets

**arXiv Math**  For the arXiv dataset, we collected a corpus of papers by downloading them via the arXiv Bulk Data Access[1]. We filtered papers to include only articles labeled as "Mathematics" and whose LaTeX source was available. The number of tokens per paper in this dataset is roughly comparable to the number of tokens per book in PG19, because LaTeX source has many special characters and the tokenizer tends to output small subwords.

**Github**  We used BigQuery[2] to obtain a large corpus of Github repositories that are published with open-source licenses. We used file endings to filter for files in the languages C, C++, Java, Python (including Jupyter notebooks), Go, and TypeScript. Individual source code files are often fairly short, and there are many dependencies and cross-references between files in the repository. To capture these dependencies, we created one long document for each Github repository by traversing the directory tree, and concatenating all of the files within it. The order in which files are traversed within the repository is random, but each subdirectory is processed as a unit, so that all the files within the subdirectory are close to each other in the resulting document. Source code is usually structured so that related files are all grouped together in the same subdirectory; this traversal preserves that structure, while still shuffling files and subdirectories in random order.

**Formal Math – Isabelle**  The Isabelle corpus consists of formal mathematical proofs of theories. We collected all 627 theories available on The Archive of Formal Proofs[3] (as of October 6, 2021) and an additional 57 theories from the Isabelle standard library[4] to create a corpus of 684 theories. All theories have open-source licenses. Each theory is a self-contained mathematical object, on topics such as foundational logic, advanced analysis, algebra, or cryptography, and consists of multiple files containing proofs. As with the Github corpus, all files that make up a theory are concatenated

---

[1] https://arxiv.com/help/bulk_data
[2] https://console.cloud.google.com/marketplace/product/github/github-repos
[3] https://www.isa-afp.org/topics.html
[4] https://isabelle.in.tum.de/

| Context | Memory | XL cache | arXiv | PG19 | C4(4K+) | GitHub | Isabelle |
|---------|--------|----------|-------|------|---------|--------|----------|
| 512 | None | None | 3.29 | 13.71 | 17.20 | 3.05 | 3.09 |
| 2048 | None | None | 2.69 | 12.37 | 14.81 | 2.22 | 2.39 |
| 512 | None | 512 | 2.67 | 12.34 | 15.38 | 2.26 | 2.46 |
| 2048 | None | 2048 | 2.42 | 11.88 | 14.03 | 2.10 | 2.16 |
| 512 | 1536 | None | 2.61 | 12.50 | 14.97 | 2.20 | 2.33 |
| 512 | 8192 | None | 2.49 | 12.29 | 14.42 | 2.09 | 2.19 |
| 512 | 8192 | 512 | 2.37 | 11.93 | 14.04 | 2.03 | 2.08 |
| 512 | 65K | 512 | 2.31 | 11.62 | 14.04 | 1.87 | 2.06 |
| 2048 | 8192 | 2048 | 2.33 | 11.84 | 13.80 | 1.98 | 2.06 |
| 2048 | 65K | 2048 | **2.26** | **11.37** | **13.64** | **1.80** | **1.99** |

Table 4: Average token-level perplexities of each model when trained for 500k steps.

together into one long document. Unlike the Github corpus, we order the files according to their import dependencies, so that later files use sub-theorems that are proved in earlier files.

**C4(4K+)**  C4, the colossal cleaned common crawl, is a very large collection of documents that have been scraped from the internet (Raffel et al., 2020). We filtered out all documents that have less than 4096 tokens to focus on documents where memory can have an impact.

**PG-19**  PG-19 is a large dataset of English-language books, published prior to 1919, which were retrieved from the Project Gutenberg archive (Rae et al., 2020; Sun et al., 2021). PG-19 is one of the few public datasets that only contains full-length books, and has become a benchmark for long-range natural language text modeling.

## 4.2 EXPERIMENTAL METHOD

We used a 12-layer decoder-only transformer (with and without Transformer-XL cache) with an embedding size of 1024, 8 attention heads of dimension 128, and an FFN hidden layer of size 4096. For all of our experiments, we used $k = 32$. Unless specified otherwise, we use the 9th layer as the $k$NN augmented attention layer. We used a sentence-piece (Kudo & Richardson, 2018) tokenizer with a vocabulary size of 32K.

We used the Adafactor optimizer (Shazeer & Stern, 2018). In preliminary experiments, we conducted a hyperparameter search to determine the optimal learning rate among three choices ($\{3.0, 1.0, 3 \cdot 10^{-1}\}$), and found that $1.0$ works best. We used a linear warmup schedule for the first 1000 steps, followed by square root decay. We trained the models from scratch for 500K steps on all the datasets, except for the Isabelle dataset. Isabelle is small, so we stopped training after 100K steps when the model began to overfit. We ran all of our experiments on 32 TPU cores. Our models were implemented in JAX (Bradbury et al., 2018) and Flax (Heek et al., 2020).

When comparing models with different context lengths, we adjusted the batch size (the number of documents in a batch) so that there are always $2^{17}$ tokens in a batch. E.g., a model with a context length of 512 has a batch size of 256, while the 2048 model has a batch size of 64.

We experimented with multiple implementations of approximate $k$NN lookup with different tradeoffs between quality and computational cost. We did not observe a significant degradation of the model quality when switching to lower quality approximations of $k$NN, so the model appears to be quite robust with respect to the quality of $k$NN retrieval. For a model with around 200M trainable parameters the step time increased from 0.2s to 0.25s when we added a memory of size 8K, and to 0.6s when we added a memory of size 65K (measured on TPUv3).

## 4.3 EFFECT OF EXTERNAL MEMORY

**Adding external memory results in substantial gains across datasets and architectures,** as shown in Table 4. Across all five datasets, adding external memory to either the vanilla Transformer or the Transformer-XL architecture improves perplexity by a substantial amount. For example, on

| Context | Pretrain | Fine-tune | Perplexity |
|---------|----------|-----------|------------|
| 512 | 8192 | None | 2.37 |
| 512 | 65K | None | 2.31 |
| 512 | 8192 | 65K | 2.32 |
| 512 | 8192 | 131K | 2.30 |
| 512 | 8192 | 262K | 2.26 |
| 2048 | 8192 | None | 2.33 |
| 2048 | 65K | None | 2.26 |
| 2048 | 65K | 131K | 2.23 |
| 2048 | 65K | 262K | **2.21** |

Table 5: Finetuning for 20K steps to make use of a larger memory on the arXiv data set.

C4(4K+) dataset, adding memory of size 8192 improves the perplexity of the vanilla Transformer (with context size 512) from 17.20 to 14.42, and improves Transformer-XL from 15.38 to 14.04.

**Increasing the size of the memory increases the benefit of the memory.** The best perplexities for all datasets and architectures were obtained with a memory size of 65K.

Note that Transformer-XL with context size 2048 already has a theoretical receptive field that is quite large. Each token in a higher layer can attend up to 2048 tokens away in the layer below, so the total receptive field is $2048 \cdot 12$ (layers) $\sim 25$K. Nevertheless, we still saw a substantial gain when adding an external memory of size 8192 to this model. $k$NN attention into memory would appear to be a more effective way to retrieve information from the distant past than the Transformer-XL cache.

On the other hand, we also saw improvements by adding XL cache to the large-memory (65K) models. In a vanilla (non-XL) Transformer, the first few tokens in a sequence have very little context, and thus have higher perplexity. The XL cache provides additional local short-range context at the start of a sequence, which complements the long-range context provided by external memory.

Interestingly, in a vanilla Transformer, using even a small external memory of size 1536 provides a gain in perplexity which is almost as good as using a local context of size 2048 but no memory (e.g. Table 4). This is surprising, because the external memory is not differentiable, and is added only to one layer of the Transformer, whereas increasing the context size is differentiable and affects all layers. We conclude that the lower layers of a Transformer don't necessarily need long-range context, and having a differentiable memory is not as important as one might suspect.

## 4.4 SCALING TO LARGER MODELS

We scaled up the Transformer model to sizes of 1 and 8 billion parameters. For the 1 billion parameter model, we use 8 layers, 32 heads with head dimension 128, $d$\_model 2048, and $d$\_ff 16384. For the 8 billion parameter model, we use 64 heads, 16 layers, $d$\_model 4096, and $d$\_ff 32768. We used a context size of 2048, memory size of 8192, and no XL cache. We ran the comparisons to the vanilla Transformer on the arXiv math dataset. Scaling plots are shown in Figure 1.

External memory provides a consistent improvement to the model as it is scaled up. Remarkably, we found that **the smaller Memorizing Transformer with just 8k tokens in memory can match the perplexity of a larger vanilla Transformer which has 5X more trainable parameters.**

## 4.5 FINETUNING ON LARGER MEMORIES

**Finetuning on a larger memory.** In some cases, training was unstable when using large memories, possibly due to distributional shift early in the training (See Section 3.2). Thus, for memories of 131K or more tokens, we first pretrain the model with a memory size of 8192 or 65K for 500K steps, and then finetune it with the larger memory for an additional 20K steps. The results of finetuning on the arXiv Math data set are shown in Table 5. **Increasing the size of external memory provided consistent gains up to a size of 262K.** Note that 262K tokens is longer than almost all of the documents in arXiv, and thus we would not expect to see any gain past this point (see Appendix A).

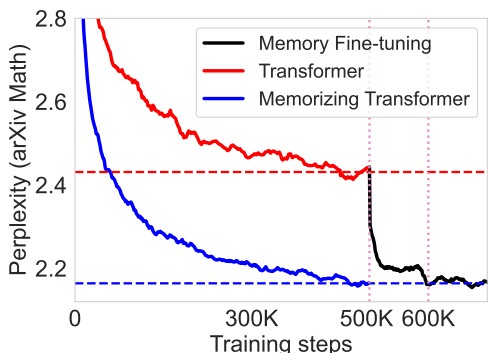

Figure 6: Finetuning a 1B vanilla Transformer model to use external memory of size 65K.

**Finetuning a non-memory model to use memory**    Pretraining can be very costly both in time and computational resources. Thus, a natural question to ask is: *can one fine-tune a pretrained Transformer to use external memory?* The answer is yes!

We took a pre-trained 1B vanilla Transformer model, and fine-tuned it to use external memory (the 1B models used in Section 4.4). The fine-tuning result is shown in Figure 6. Notice that the model quickly learns to use external memory. Within 20K steps (4% of the pre-training time) the fine-tuned model has already closed 85% of the gap between it and the 1B Memorizing Transformer, and after 100k steps it has closed the gap entirely.

## 4.6 INFORMATION RETRIEVAL PATTERNS

We conducted a qualitative study of what the model was actually retrieving from external memory, by finding which tokens showed the biggest improvements in cross-entropy loss when the size of the memory was increased, and then examining the top-$k$ retrieved memories for those tokens. We found that the model gained the most when looking up rare words, such as proper names, references, citations, and function names, where the first use of a name is too far away from subsequent uses to fit in the local context. This result is in keeping with the prior analysis of long-context Transformers on PG19 (Sun et al., 2021), which found similar lookup patterns. For this experiment, we used a slightly older version of the architecture without the gating mechanism.

**Which tokens show a benefit from memory?**    Figure 7 shows a visualization of which tokens show an improvement when the size of the external memory is increased. We selected a math paper at random, and plotted the difference in cross entropy loss for each token $x_i$ in the paper, comparing two models with the same parameters, but with memories of different sizes. $\Delta_i = \text{cross-entropy}_{8192}(x_i) - \text{cross-entropy}_{32K}(x_i)$. Positive values show an improvement in loss.

The $x$-axis on the chart is the token number $i$, while the $y$-axis is $\Delta_i$. For the first 8192 tokens, the difference between the two models is zero, since the larger capacity of the 32K memory isn't being used yet. However, after token 8193, we can see that the larger memory helps, on average, over the smaller memory. The benefit is not universal, since the predictions for some tokens become worse, possibly due to the fact that a relevant retrieved memory no longer makes it into the top-$k$ when the size of the external memory is increased. This figure also shows that the benefit of external memory is somewhat sparse. The improvement in perplexity seems to be mainly driven by a small percentage of tokens that obtain a large improvement in cross-entropy loss when using the larger memory.

**What information is being looked up?**    Given that only a subset of tokens shows improvement from external memory, we did a further investigation into what, exactly, those tokens are using the memory for. We took those tokens which showed the largest improvement in cross-entropy loss, and for each of them tokens, we examined the top-$k$ retrieved memories. We studied arXiv math, Github and Isabelle corpus. For arXiv math and Github, we found the model retrieved function and variable names. See more details with examples in Appendix B.

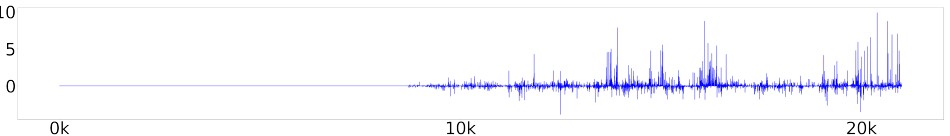

Figure 7: Difference in loss for each token in a randomly chosen paper, using the same model once with a memory size of 8K and once with 32K. Higher numbers mean the longer memory helped in comparison to the shorter memory. This paper is 22K tokens long.

| Query index | Input | Target | Surrounding context | Retrieved index | Retrieved surrounding context |
|---|---|---|---|---|---|
| 29721 | mark | ov | rule prob_space. markov_inequality | 8088 | M. t \<le> X a} \<le> expectation X / t" |
| 40919 | _ | th | = ( subgraph_threshold H n / p n) | 27219 | threshold H n = n powr (-(1 / max_density' |
| 49699 | S | w | assumes " orthonormal_system S w" | 28050 | definition orthonormal_system :: " |

Table 8: Examples of memory retrieval in the Isabelle dataset. The model is able to find the definition of a lemma from a reference to it. The retrieved surrounding context (highlighted) is the definition body of the mathematical object highlighted in the querying context.

*Retrieving mathematical definitions.* Our case study on the Isabelle corpus provides one of the clearest illustrations of how a model can make good use of external memory. When predicting the name of a mathematical object or a lemma, the model looked up the definition from earlier in the proof. Examples of this behavior are shown in Table 8. In example 1, the model retrieves a definition within the body of a lemma, `markov_inequality`. In example 2, it retrieves the definition of a previously defined concept `subgraph_threshold`. In example 3, it retrieves the definition of `orthonormal_system`. We manually checked 10 examples where the model made a prediction of lemma names, and 8 out of 10 times model found the body of the lemma it needs to predict. In the other two cases, the model also looked up materials in the immediate vicinity. To the best of our knowledge, this is the first demonstration that attention is capable of looking up definitions and function bodies from a large corpus. The Isabelle case study used a model with two memory layers of size 32K.

## 5 CONCLUSION

We present a simple extension to the Transformer architecture, called $k$NN-augmented attention, which dramatically increases the length of the context that a language model can attend to by using $k$-nearest-neighbor lookup into a large external memory. We demonstrate the effectiveness of external memory in a series of language modeling experiments over a variety of long-document datasets, including LaTeX documents, source code, formal proofs, and books.

The Memorizing Transformer shows large improvements in perplexity over the baseline for all of the data sets and architectures that we studied; it is comparable to a vanilla transformer that has 5 times the number of parameters. Perplexity continues to improve with increasing memory size, although there is a point of diminishing returns. Moreover, external memory continues to provide benefits even as the transformer is scaled up from 200M to 8B parameters. Perhaps most intriguingly, a Memorizing Transformer does not need to be pre-trained from scratch; it is possible obtain large gains from adding memory to an existing pre-trained model, and then fine-tuning it.

Unlike other forms of attention, $k$NN retrieval can be easily scaled up to huge memory sizes, and is thus potentially able to leverage vast knowledge bases or code repositories. How to make the best use of this capability is a topic for future work.

ACKNOWLEDGMENTS

We want to thank Charles Staats for the many fruitful discussions and detailed comments, Henryk Michalewski for early version of of the memory implementation, Petros Maniatis for his help with our code datasets, Aitor Lewkowycz for his help with larger scale memorizing transformer experiments, Behnam Neyshabur for his comments on finetuning non-memory models, Imanol Schlag for his proofread and detailed comments, and Dennis Lee and Manzil Zaheer for discussions about large-scale attention and retrieval.

## ETHICS

The ability to memorize large databases of facts could have potential ramifications for society, especially if those databases include sensitive personal information or copyrighted works. However, one advantage of using an external memory is that the memory can be easily cleared of all such information, as we do at the end of each document that we train on. The same is not true of differentiable model parameters, which is what most existing architectures use to store facts and information that they are trained on.

## REPRODUCIBILITY

Details of our architecture and training hyperparameters are given in Section 4.2. The datasets for C4 and PG-19 are publicly available. Our additional datasets, Github, Isabelle, and ArXiv Math are derived from publicly available data buckets, which we link in the main part of the paper. Subsection 4.1 include details on how we constructed the datasets from those datasets. We plan to release our code as open source.

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

## A    LENGTH OF INPUTS

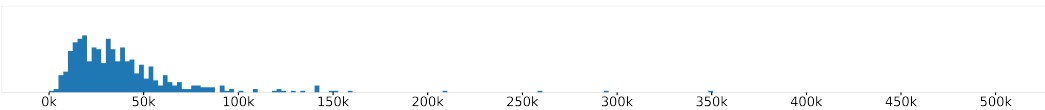

Figure 9: Histogram of the number of tokens in arXiv math papers dataset. We tuncated the histogram at 500k tokens. The maximum paper had almost 1.6M tokens.

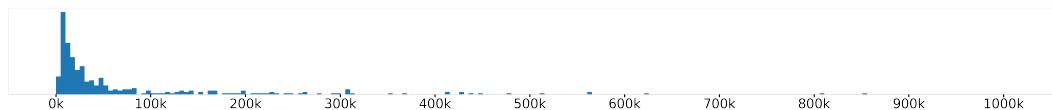

Figure 10: Histogram of the number of tokens in Github repositories dataset. We cut off the long tail of this plot. The repository with the maximum length has just over 9M tokens.

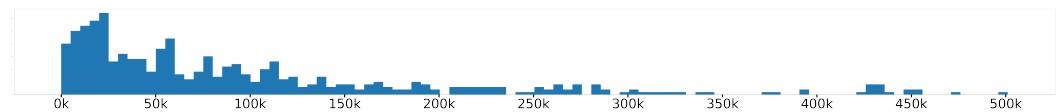

Figure 11: Histogram of the number of tokens in Isabelle proof scripts dataset.

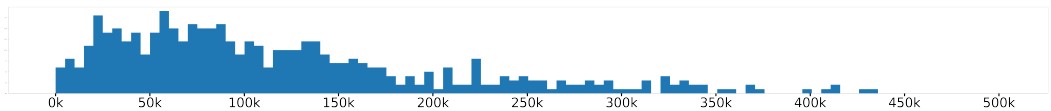

Figure 12: Histogram of the number of tokens in PG19 books dataset.

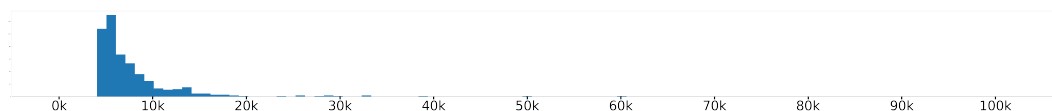

Figure 13: Histogram of the number of tokens in C4 documents filtered by documents that have less than 4096 tokens.

## A.1 ABLATION STUDIES

In the following section, we performed ablation studies to investigate the effects of various hyperparameters. Unless otherwise specified, we carried out these experiments with a memorizing transformer with context size 512, XL cache 512 with a memory size of 8192.

**Multiple $k$NN layers.** We experimented with using two $k$NN layers, rather than just one. However, we did not see further benefits brought by more than multiple retrieval layers.

$k$**NN layer index** We experimented with adding the external memory to layer 3, 6, 9 and 12 in a 12-layer transformer, with results shown in Table 14. We found that adding memory to the middle of the layer stack will obtain the best result, whereas adding memory to layers either too close to the input or to the output obtained less gains.

Table 14: Different layer index.

| Layer index | Perplexity |
| --- | --- |
| 3 | 2.40 |
| 6 | 2.36 |
| 9 | 2.37 |
| 12 | 2.43 |

**Number of neighbors** We studied the effects of the number of neighbors we retrieve from memory, with results shown in Table 15. We found that even with 32 number of neighbors, we can already obtain a comparable results with 128 or 256 neighbors.

Table 15: Number of neighbors.

| Number of neighbors | Perplexity |
| --- | --- |
| 32 | 2.38 |
| 128 | 2.37 |
| 256 | 2.37 |

**Random seeds** We measured the statistical significant of the results reported. We did 3 runs with 3 random seeds for Transformer XL of size 512, and also a memorizing transformer with memory size 8192. We measured the standard deviation of perplexities after 500K steps of training, shown in Table 16. We saw the standard deviation between different runs of the same experiment appears to be much smaller than the gap between different models.

Table 16: Random seeds.

| Models | Perplexity |
| --- | --- |
| Transformer XL | $2.67 \pm 0.01$ |
| Memorizing Transformer | $2.37 \pm 0.005$ |

## B WHAT DOES THE MODEL RETRIEVE FROM MEMORY?

**Retrieving citation names**  On arXiv math, several examples are shown in Table 17, which includes both the retrieved token and its surrounding context. We observe that many of the gains in cross-entropy loss took place when trying to predict the name of bibitems, citations, or references, by looking up the references and citations used previously in the paper. Such lookups usually span over the entire paper, which is much longer than 8192 tokens, providing a plausible explanation for the gain beyond memory size of 8192.

Table 17: The table shows several examples of which tokens were retrieved during language modelling of arXiv math dataset. The model is retrieving names of the references from previous passages.

| Query index | Input | Target | Surrounding context | Retrieved index | Retrieved surrounding context |
|---|---|---|---|---|---|
| 20389 | Mon | thus | bibitem{ ComtetMonthusYor } | 2208 | Brownian motion \cite{ ComtetMonthusYor } |
| 16623 | cha | kra | \cite{ chakrabarti }. | 4677 | ∼1.2 of \cite{ chakrabarti } |
| 14747 | as | d | \eqref{ asdfg } which | 3365 | begin{equation} \n \label{ asdfg .1} |

**Retrieving function names from the codebase**  As with the arXiv papers, we also studied which tokens the model retrieved from memory. As might be expected, the model is often looking up the names of functions, and variables, as shown in Table 18.

Table 18: Examples of memory retrieval in the Github dataset. The model looks up how functions are used elsewhere in the repository.

| Query index | Input | Target | Surrounding context | Retrieved index | Retrieved surrounding context |
|---|---|---|---|---|---|
| 23837 | Fo | nte | menu_play-> setarFonte | 14607 | menu_load-> setarFonte |
| 23825 | , | 35 | hscreen/2-50, 50, 200, 35 ); | 14599 | 20, y+40, 200, 35 ) |
| 14546 | -> | adi | panel-> adicionaComponente | 5205 | panel-> adicionaComponente |

## B.1 MORE RETRIEVING EXAMPLES IN FORMAL THEOREM PROVING CORPUS

**Example 1**

- Input token index: 64604
- Input token: "_"
- Target token: "pair"
- Surrounding context: )) by (simp add: Fourier_sum_limit_pair [OF f, symmetric] Fourier'
- Name needs to be predicted: `Fourier_sum_limit_pair`
- Retrieved token: "Four"
- Retrieved token index: 64412
- Retrieved context: 2 * n. Fourier_coefficient f k * trigonometric_set k t)
- Definition of the name:

```
lemma Fourier_sum_limit_pair:
  assumes "f absolutely_integrable_on {-pi..pi}"
  shows "(λn. ∑k≤2 * n. Fourier_coefficient f k * trigonometric_set k t) ⟶ l
    ⟷ (λn. ∑k≤n. Fourier_coefficient f k * trigonometric_set k t) ⟶ l"
      (is "?lhs = ?rhs")
```

Figure 19: Definition of `Fourier_sum_limit_pair`.

**Example 2**

- Input token index: 46175
- Input token: "tri"'
- Target token: "gon"
- Surrounding context: <le>n. a k * trigonometric_set k x)
- Name needs to be predicted: `orthonormal_system_trigonometric_set`
- Retrieved token: "gon"
- Retrieved token index: 35457
- Retrieved context: lemma orthonormal_system_trigonometric_set:\n "orthonormal_system
- Definition of the name:

```
lemma orthonormal_system_trigonometric_set:
    "orthonormal_system {-pi..pi} trigonometric_set"
```

Figure 20: Definition of `orthonormal_system_trigonometric_set`.

## Example 3

- Input token index: 49760
- Input token: "sum'"
- Target token: "m"
- Surrounding context: nusing Fourier_series_square_summable [OF assms, of'
- Name needs to be predicted: `Fourier_series_square_summable`
- Retrieved token: "sum"
- Retrieved token index: 35457
- Retrieved context: lemma Fourier_series_square_summable\n assumes:
- Definition of the name:

```
lemma Fourier_series_square_summable:
  assumes os: "orthonormal_system S w" and w: "⋀i. (w i) square_integrable S"
    and f: "f square_integrable S"
  shows "summable (confine (λi. (orthonormal_coeff S w f i) ^ 2) I)"
```

Figure 21: Definition of `Fourier_series_square_summable`.

## Example 4

- Input token index: 49697
- Input token: "_'"
- Target token: "system"
- Surrounding context: lemma Riemann_lebesgue_square_integrable: nassumes "orthonormal_system S w
- Name needs to be predicted: `orthonormal_system`
- Retrieved token: "system"
- Retrieved token index: 28052
- Retrieved context: definition orthonormal_system :: "\'a::euclidean'
- Definition of the name:

```
definition orthonormal_system :: "'a::euclidean_space set ⇒ ('b ⇒ 'a ⇒ real) ⇒ bool"
  where "orthonormal_system S w ≡ ∀m n. l2product S (w m) (w n) = (if m = n then 1 else 0)"
```

Figure 22: Definition of `orthonormal_system`.

**Example 5**

- Input token index: 34817
- Input token: ".""
- Target token: "b"
- Surrounding context: shows "integrable (lebesgue_on {a..b})
- Retrieved token 1: "."
- Retrieved token index 1: 2416
- Retrieved context 1: lebesgue_on {a..b}) f i
- Retrieved token 2: "-"
- Retrieved token index 2: 2445
- Retrieved context 2: (lebesgue_on {a-c..b-c}) (
- Retrieved token 3: "-"
- Retreived token index 3: 6479
- Retrieved context 3: (lebesgue_on {-pi..pi}) (

**Example 6**

- Input token index: 49759
- Input token: "_""
- Target token: "sum"
- Surrounding context: 0"\n using Fourier_series_square_summable [OF assms
- Retrieved token 1: "set"
- Retrieved token index 1: 35044
- Retrieved context 1: definition trigonometric_set :: "nat \<Rightarrow>
- Retrieved token 2: "ier"
- Retrieved token index 2: 47272
- Retrieved context 2: definition Fourier_coefficient\nwhere
- Retrieved token 3: "ine"
- Retrieved token index 3: 18160
- Retrieved context 3: lemma Schwartz_inequality_strong:\nassumes "f'
- Retrieved token 4: "system"
- Retrieved token index 4: 28052
- Retrieved context 4: definition orthonormal_system :: "\'a::euclidean'
- Retrieved token 5: "<"
- Retrieved token index 5: 47241
- Retrieved context 5: subsection\<open>Convergence wrt the L'
- Retrieved token 6: "n"
- Retrieved token index 6: 40835
- Retrieved context 6: \n subsection\<open>A bit of extra'

