# OpenReview forum: "Memorizing Transformers"
_ICLR.cc/2022/Conference — ICLR 2022 Spotlight_

### Official Review · Reviewer_75xx · 2021-10-27

**Correctness:** 3
**Technical Novelty And Significance:** 4
**Empirical Novelty And Significance:** 3
**Recommendation:** 8
**Confidence:** 5

**Main Review:**

Strengths:

1. The method proposed is novel, intuitive and simple to implement. This is a huge advantage in the language modeling space. We’ve seen many memory-augmented transformers in the past few years, and most of them have failed to have significant impact because they are too complicated to replicate or take too many resources to train.
2. The results on the C4 dataset seem significant and show that  memory augmentation could really improve transformer language modeling.
3. The analysis is super interesting, and it shows the potential of this simple approximate attention method to find very relevant contexts to retrieve from the recent past, and how these retrievals aid in improving the LM.
4. The paper is well-written and easy to understand.

Weaknesses:

1. The authors do not compare their model to any previously published model. Throughout the paper the authors only compare their model to baselines that they designed and trained. Doing this for some of the datasets is fine, but not even having one dataset in which you compare to strong, previously published models makes it harder to convince the reader that you have found something interesting. You did show results for PG19, which has been used before in LMing papers, so it’s not clear why you haven’t compared to previously published models on that dataset, such as the compressive transformer? The numbers in the compressive transformer are very different from the numbers your models obtain, so I’m assuming that you preprocessed that dataset differently and so have a different vocabulary, making those results incomparable? Could you re-train your model using the same settings as the compressive transformer so that we could compare the results of your method and theirs?
2. The baseline has just 6 layers, which is significantly smaller than the current transformer LMs that are being discussed by the community. I’m definitely not expecting to see GPT-3 sized models in every research paper (or even close to that) but I think 16 layers would make more sense for such an exploration. I’m afraid the gains shown here would not transfer to larger models. On the other hand, larger models might produce better representations resulting in even bigger gains.
3. The authors do not show standard deviation numbers for any of the baselines on the various datasets. This makes understanding the results on each dataset, for the different models, harder. For example, on the arxiv dataset the authors talk about a drop from 2.927 to 2.877 perplexity between two different models, but is a 0.05 perplexity difference really statistically significant here?
4. The authors don’t provide any speed or memory usage statistics. Of course, it’s totally fine if this model runs slower or uses more memory than the baseline- I expect such preliminary investigations to potentially be nonoptimal, with later work making it an actually viable model. I would not penalize this paper for having slow runtime or using too much memory. But I think it's important to discuss these things.


Misc:

1.  Although the perplexity improvements on many of the datasets are quite small (<=.1 perplexity) this could be a non-issue, and more of a problem with the perplexity metric than with the model. The perplexity metric, and most of the data that we train our LMs on, heavily incentivizes models to get short term dependencies right. Getting the model to look back into the past is both hard and barely incentivized, and so these small initial gains that are attained in this paper might be improved when we find a better loss function that incentivizes looking back more, or when we find data that has more long-term dependencies and less short-term ones.
2. The way I see it, there are two different ways to augment the transformer with a memory module. The memory module could be ‘global’ or ‘local’. Local memory transformers have a memory module that can only access the recent past. So models like the compressive transformer or Transformer-XL are local memory transformers. On the other hand, models like the kNN-LM or REALM are ‘global’ memory models, since they can access the entire datastore for every prediction, without any kind of limitation that is dependent on the current timestep and the recent previous timesteps.
The model proposed here is clearly a ‘local’ memory transformer, since it performs approximate attention over the recent past. I believe that that is part of what makes this model work, and I think the paper might benefit from a discussion of this and maybe even an empirical test. Does this model improve performance on WikiText-103, where all the articles are in a shuffled ordering? I don’t think it would, and that’s totally fine, but I think readers would benefit from a discussion of this property of the model.
3. I think it could be helpful if the authors had more comparisons where their model and the baseline can access the same amount of tokens for every prediction. For example, if you train one of the memory models with 2k tokens of memory and 512 tokens of local context it would be interesting to compare it to the baseline model trained on 2.5k tokens. Of course, since your model performs approximate attention we do expect to see worse perplexity values, but it would be very insightful to see precisely how much using this approximate attention method degrades performance.
4. For section 3.2, consider referencing [CrossBatch] which discusses this distributional shift in section 3.2 of their paper, in the vision domain.

References:

[Compressive Transformer] Jack W. Rae, Anna Potapenko, Siddhant M. Jayakumar, Timothy P. Lillicrap. “Compressive Transformers for Long-Range Sequence Modelling”

[REALM] Kelvin Guu, Kenton Lee, Zora Tung, Panupong Pasupat, Ming-Wei Chang. “REALM: Retrieval-Augmented Language Model Pre-Training”

[CrossBatch] Xun Wang, Haozhi Zhang, Weilin Huang, Matthew R. Scott. “Cross-Batch Memory for Embedding Learning”


**Summary Of The Paper:**


Transformer language models attend to the previous N tokens, and so do not have any access or knowledge of the tokens that appeared more than N tokens ago. Extending N would allow language models to have more knowledge which might increase performance. But there’s a tight limit as to how far we can naively extend N, since memory usage grows quadratically with N, and so in SoTA transformer LMs N is limited to a few thousand tokens.

This paper shows a novel, simple and intuitive method for extending the number of tokens a transformer can access at every timestep. The authors augment the model with a kNN datastore, so that in the top layer, the model attends to the local context using normal attention but then also attends to up to 262k previous tokens by using an approximate nearest-neighbors datastore. This is basically equivalent to approximately attending over those previous tokens. No gradients are passed to the representations in the datastore in any way, making this easy to implement and relatively efficient to train.

The authors present results on multiple language modeling datasets that have long-range dependencies. On most of the datasets the gains from using the proposed method are quite small (<=.1 perplexity), and it’s not clear that they are statistically significant, but on the C4 dataset the model reduces perplexity by 2 which I believe to be a strong result. I believe that this idea is promising and that giving it a stage at ICLR will push the community to try further improving upon it, hopefully leading to larger gains.



**Summary Of The Review:**

This paper introduces a new method for extending the transformer model with a simple datastore. The insert and query operations into this datastore do not have to be trained which is a big advantage for efficiency and simplifies the implementation. Results are shown on multiple datasets, with the model showing big gains on one and smaller gains on the others. I believe the idea presented is reproducible, and extensible and I’m looking forward to seeing how this research direction develops. The analysis presented is insightful and could guide further research. I believe this paper should be accepted.

---

> ### Author Response · Authors · 2021-11-18
> **Response to 75xx**
>
> Thank you for the detailed feedback and insights.
>
> *Baselines and Larger Models:* We fully agree the paper would profit from additional baselines and experiments with larger models (more layers). We are adding comprehensive experiments using Transformer-XL with 12 layers. For details please check the general response. Additionally, we ran experiments with a model with 1B parameters, where the addition of memory also helped significantly. We will report those in the final version of the paper.
>
> *Speed and memory usage:* See common response
>
> *Standard deviation:* See common response.
>
> *“On most of the datasets the gains from using the proposed method are quite small (<=.1 perplexity)”* We fully agree with the reviewer's point on the perplexity metric.  Perplexity is computed as an average over all tokens, and especially for natural language (PG19, C4), the majority of tokens do not necessarily require much long-range context in order to make accurate predictions.  Thus, adding external memory may dramatically improve the predictions for a few tokens (as we found in our case study), while making only small improvements to overall perplexity.
>
> We also would like to point out that the absolute perplexity gain is not a very meaningful measure, as it depends heavily on the tokenization.  As we mention above, datasets with many single-character tokens (e.g. arXiv, Github, Isabelle) will tend have lower perplexity, simply because predicting a single character is only selecting from a small subset of the total vocabulary.  If the perplexity is already low, small improvements can be significant.
>
> For example, one can see that in compressive transformer, the gain of the model on Enwiki8 is from 0.98 (transformer-XL) to 0.97 (compressive transformer) in terms of bits-per-character, which corresponds to a per-character perplexity improvement from 1.972 to 1.959.  This improvement is considered as significant on that dataset.
>
> *“global vs local memory model”:* This is a very intriguing point. The ambition of our research is indeed to explore 'global' memory models. To that extent we have already performed a preliminary experiment on the English to German translation task, where we concatenate many (randomly shuffled) translation pairs into long strings. While the benefits of the memorizing transformer were less pronounced than in long-document language modeling, we did observe some improvements. This seems to suggest that the model can benefit not only by retrieving relevant context from the recent past, but also by querying the dataset itself.
>
> *"2k tokens of memory and 512 tokens of local context vs 2.5k tokens of local context":* Great idea - we will add this experiment to the final version of the paper.
>
> *CrossBatch:* Thanks for the reference, we will study the paper and reference it in the final version.

---

> > ### Author Response · Authors · 2021-11-18
> > **Followup on our response.**
> >
> > Hi reviewer 75xx, as the interactive discussion period is ending soon, we appreciate your time spent reviewing our manuscript and would be grateful if you can confirm whether our response addressed some of the concerns raised in the review? Please let us know if any issues remain and/or if there are any additional clarifications we can provide.

---

> > > ### Comment · Reviewer_75xx · 2021-11-19
> > > **response**
> > >
> > > Hi!
> > > > "absolute perplexity gain is not a very meaningful measure"
> > >
> > > You are totally correct. I'm used to seeing results on datasets that have larger standard deviation so the absolute differences here seemed small but now that you've added the STD it makes everything more clear!
> > >
> > > Overall I believe the ideas in this paper are simple and interesting and I strongly recommend accepting this work.

---

> > ### Public Comment · ~Hongyuan_Mei1 · 2022-04-30
> > **code release timeline; more details about tokenization and what "context" means**
> >
> > Dear authors,
> >
> > This is really great work and I was excited reading it!
> > When would you plan to release the source code?
> >
> > I tried to reproduce the model on my own but can not achieve a comparable PPL.
> > I suspect tokenization and attention sliding window are the source of differences.
> >
> > Could you please provide more information about the tokenizer?
> > In particularly, what is it trained on; what is the frequency threshold; how many tokens (train/valid/test) did you end up with after tokenizing data; what is the renormalized word-level perplexity?
> >
> > Could you also please clarify whether the first 2 rows in Table-4 still has the trick of "use a sliding-window causal mask (Beltagy et al., 2020) so that each token has a local context that includes the previous 512 tokens"?
> >
> > Thank you very much!
> >
> > Best,

---

> > > ### Public Comment · ~Markus_Norman_Rabe1 · 2022-06-22
> > > **Source code**
> > >
> > > Dear Hongyuan, we released the source code as part of the meliad repository: [https://github.com/google-research/meliad](https://github.com/google-research/meliad).

---

### Official Review · Reviewer_zRRc · 2021-10-27

**Correctness:** 4
**Technical Novelty And Significance:** 2
**Empirical Novelty And Significance:** 2
**Recommendation:** 6
**Confidence:** 4

**Main Review:**

Pros:
- propose a non-differentiable cache mechanism to reuse keys and values coming from prior training steps.
- comprehensive experiments are conducted to evaluate the effectiveness of the proposed memory-based method.
Cons:
- there are some benchmarks that require models to be able to handle long inputs, such as document-level QA datasets and document-level retrieval. Besides the LM task, it could be better if more document-level downstream tasks could be evaluated to verify the usefulness of the proposed approach.


**Summary Of The Paper:**

This paper presents a memory-based Transformer where a memory is used to leverage the external inputs for scenarios that have long inputs like documents or codes. The key contribution is that it describes how external contexts are integrated into the representations of the current inputs. By introducing such external contexts with the memory mechanism, perplexity of LM can be improved on every dataset studied in the paper.

**Summary Of The Review:**

This work is highly related to those scenarios, such as document-level QA/retrieval and code completion, where the models should consider and leverage long inputs. The key contribution, based on my understanding, is the joint KNN/local attention part, where both external context and local context are combined. In general, the paper is well written and all details are clear to me. One thing could be further improved is that I suggest to include more downstream tasks to verify the effectiveness of the proposed method.

---

> ### Author Response · Authors · 2021-11-19
> **Response to zRRc**
>
> Thank you for the feedback.
>
> *“More document-level downstream tasks could be evaluated to verify the usefulness of the proposed approach.”*
>
> We agree that additional evaluation would be helpful, but we would like to argue that language modeling already is a very diverse task. For example, the GPT papers have shown that language modeling alone equips models with a wide range of abilities. Instead of focussing on downstream tasks, we chose to demonstrate improvements across several datasets, which differ greatly in the kind of data they contain.

---

> > ### Comment · Reviewer_75xx · 2021-11-19
> > **Strongly agree with the authors**
> >
> > I would like to very strongly state that I fully agree with the authors here. Language modeling papers that do __just__ language modeling are super interesting and should not be rejected or penalized for not having results on downstream tasks. The authors need to focus on a clear and sharp contribution, and in most cases page limits, deadlines and amount of authors and compute resources limit the scope of a paper.
> >
> > Of course every paper would be stronger if it had more experiments, but we know how hard it is to write a paper, so we should be very conservative and understanding with what we ask authors to present.

---

### Official Review · Reviewer_eMDi · 2021-10-29

**Correctness:** 3
**Technical Novelty And Significance:** 2
**Empirical Novelty And Significance:** 2
**Recommendation:** 5
**Confidence:** 4

**Details Of Ethics Concerns:**

Some of the experiments in the paper were implemented on data downloaded online.
This data includes scientific papers from arXiv, code from github, and mathematical proofs from another repository. It is unclear if the authors have rights to apply their work on these papers.
Taking code for a dataset requires proper licenses, and same goes for arXiv papers that may have different licenses (different CC types, etc).
The authors don't mention how they deal with copyrights.


-----------------
Update:

Thanks to the authors for clarifying how they deal with these aspects.




**Main Review:**

The paper proposes a novel method to connect a k-NN based memory with transformers for language modeling. The idea seems interesting and somewhat novel and evaluated empirically. The previous work is nicely identified, although the work of Rae et al. (ICLR 2020) could be described there as well.

Strong points:
- The memory seems to be usefully working and having a positive effect.
- Practically, seems to work.
- Relevant work to this community.

Weak points:
- Section 3 could see more formalization and better description on what are the input for searching the kNN, how the model is trained (maybe, an algorithm figure).
- Experimentation could be greatly improved: no hyperparameter search or reasoning behind the selection, no baselines with other memory architectures showing differences either performance or specific use-cases (beyond plain transformers).
- Memory sizes are too big for the data selected. It is unclear why such big memories are still improving the model.
- The results on PG-19 doesn't seem to replicate previous work (original paper shows 33.6 for Compressive transformer, whereas this work shows ~19 for plain transformers).
- Results are very close for each memory size and there are no error measurements (e.g., std dev). It is hard to conclude anything about memory sizes and their effectiveness.
- Hard to replicate: several proposed datasets are downloaded from the web, it will be hard to download the same files.
- No details on the specific hardware and software used.

Minor:
- What do you refer with "16 attention heads of size 64"?
- Describe the content of the figures in the caption (e.g., what is the architecture about?)

**Summary Of The Paper:**

This work proposes to deal with long-range dependencies in sequences using a transformer language model augmented with memory. The memory comes in the form of a k-Nearest Neighbor (kNN). The memory stores the last M  (Key, Values) from the previous-to-last layer, and proposes k neighbors from the memory per element in the sequence to attend to the last layer, in addition to the current ones in the sequence. The last layer is trained to utilize the memories. The memory is non-differentiable, but it is still possible to be updated, and used an "input" to a layer in the network. The authors test their model with math paper from arXiv (in Latex), PG-19, C4, Github code, and Isabelle (theorem proving).
The experiments measure performance of the language models with perplexity. The context (input) length is typically fixed to a relatively small number (512). The paper studies effects of memory vs no-memory,  different sizes of memory, and fine-tuning by training on small memory first followed by an increase in memory size. An alternative with 2 k-NN memories for the last 2 layers was tested too. Further, a few examples show how the memory is most usefully used (mostly for predicting rare tokens). The experiments show improvements in perplexity when using memory, and some saturation points.

**Summary Of The Review:**

This is an empirical work that aims at improving the processing of long-range dependencies in sequential data.
I've updated my score  recommendation is to accept the paper given that the additional experiments and standard deviations are finalized and included in the paper. I would recommend to further include a word-level tokenization of the PG-19 dataset (and maybe others) to make them comparable to previous work [Rae et al.] and more challenging to the model. This would allow readers understand whether such benefits are indeed beyond the compressive transformer or other work. Also, as the authors are using character-level language modeling, it would be much better to use the bits-per-character metric instead of perplexity.

My recommendation is to reject this work given the poor quality of the experimental section (see above and below). Also, there will be some challenges to reproduce this work with respect to the datasets that should be addressed as well.

Questions:
- Please, could you clarify how did you choose the hyperparameters and justify them?
- What is being used as search in k-NN? The output of the layer connected to the memory? Are the keys used in the search or both key/values?
- Why did you choose k=128 in k-NN? What would happen if these value changes?
- Why do you use one layer of memory, if two improve the results?
- Could you add std dev. values to all your experiments?
- Please explain why does the arXiv dataset have such a low perplexity?
- How many sequences are used for training, validation and testing?

==================================

Update:

I've updated my score based on the rebuttal discussion with the authors, and the additional elements presented. The new score is based on the existing merit, however, the contributions are limited and the experimental results require further work. The additional experiments could strongly benefit the understandings of the benefits of this method. For example, finalizing the experiment with 2K vs 500 + 1.5K memory with transformer and transformer-XL, would add lots of value. The presented results are difficult to compare with previous work (other than the included transformer-XL) due to the selected tokenizer. Therefore, including other models, e.g., compressive transformer, is missing. This could have been avoided by reproducing the experiments from those works.

---

> ### Author Response · Authors · 2021-11-18
> **Response to Reviewer eMDi**
>
> Thank you for the detailed suggestions on how to improve the paper. We are addressing the points and will upload a revised version soon.
>
> *“Could you clarify how did you choose the hyperparameters and justify them?”* See "Hyperparameters" in the common response.
>
> *"What is being used as search in k-NN? The output of the layer connected to the memory? Are the keys used in the search or both key/values?"*  We will try to clarify the description in Subsection 3.1. The local context contains keys and values from tokens for the current training step, while the external memory contains keys and values for tokens from previous training steps. The kNN-augmented layer combines the results of attention over both local context and external memory -- for each query in the current set of tokens, it returns a weighted sum of values from both the local context, and from external memory.
>
> *"Why did you choose k=128 in k-NN?"*: In preliminary experiments, we determined that k=128 is a safe approximation of attention.  Lower values of k may reduce the accuracy of the approximation, while higher values can become quite expensive.  We plan to include an ablation with k=32 in the final version.
>
> *"Why do you use one layer of memory, if two improve the results?"*: Thank you for raising this point. There is a point of diminishing returns, and the benefits of adding additional retrieval layers do not necessarily justify the memory and computational costs.
>
> *"Add std dev. values"*: See “statistical significance” in the common response.
>
> *"Please explain why does the arXiv dataset have such a low perplexity?"*: We believe this is primarily the result of tokenization.  LaTeX source code includes many special characters that are tokenized individually, especially for math papers, so their perplexity per-token is closer to what a character-level sequence model might produce, rather than a typical word-level model.  (Character-level models have a much smaller vocabulary, and thus a lower perplexity.)   Note that other datasets which use lots of special characters, such as code (Github) or theorem proving (Isabelle) similarly have lower perplexity than of natural language (C4, PG-19).
>
> *"How many sequences are used for training, validation and testing?"* PG-19, for example, includes 28,752 books (sequences), of which 50 are used for validation and 100 are used for testing. We will add detailed statistics about the number of tokens and sequences in each dataset to the Appendix. Except for the Isabelle dataset, the number of sequences was large enough so that overfitting was not an issue.
>
> *Experiments section:* We fully agree that the paper would profit from additional experiments as you suggested. Please find additional baselines and experiments in the common response.
>
> *"Memory sizes are too big for the data selected."* All our datasets contain some very long documents (see Appendix A). Note that the very long documents contribute a disproportionate amount of tokens to the dataset and therefore may have a larger impact on the perplexity than one might expect.
>
> *"Hard to replicate."* For easy replication of the results, we included C4 and PG-19, two widely available datasets. We believe the results on the other datasets can be replicated approximately with using publicly available data sources.
>
> *"No details on the specific hardware and software used."* We used two hardware setups: 32 TPU chips and 256 TPU chips.  Our models are implemented in Jax and Flax, starting from the Flax WMT transformer example (and for our new Transformer-XL baseline we started from the T5 codebase). We will add a detailed description of the setup to the paper and plan to release the code in the future.
>
> *"What do you refer with '16 attention heads of size 64'?"* Each of the 16 attention heads uses query, key, and value vectors of dimension 64.
>
> *"the work of Rae et al. (ICLR 2020) could be described"*: Thanks for noticing this oversight. We are now discussing their work in the related work section.

---

> > ### Author Response · Authors · 2021-11-18
> > **Followup on our response.**
> >
> > Hi reviewer eMDi, as the interactive discussion period is ending soon, we appreciate your time spent reviewing our manuscript and would be grateful if you can confirm whether our response addressed some of the concerns raised in the review? Please let us know if any issues remain and/or if there are any additional clarifications we can provide.

---

> > ### Comment · Reviewer_eMDi · 2021-11-19
> > **re: Reponse**
> >
> > Thank you for taking time to respond to my questions. I also took time to read the other reviews and responses.
> >
> > I would like to see some of your answers in the paper before the end of the discussion. This is a unique opportunity you have during the ICLR review.
> >
> > 1. Quick follow up on Section 3.1: How do you define a vector in the k-NN memory for later find the neighbors? Is it a concatenation of the key and value, or only one of them?
> >
> > 2. You haven't addressed my comment before: "The results on PG-19 doesn't seem to replicate previous work (original paper shows 33.6 for Compressive transformer, whereas this work shows ~19 for plain transformers)."
> > Why is this gap between your paper and Rae et al. in the reporting?
> >
> > 3. Please, could you report the results on the "2k tokens of memory and 512 tokens of local context vs 2.5k tokens of local context" or similar experiment before the end of the discussion period?

---

> > > ### Author Response · Authors · 2021-11-23
> > > **re: Response**
> > >
> > > 1. The memory contains (key, value) pairs. The distance measure used for kNN is the negative dot product between query and key. kNN then returns the (key, value) pairs with the lowest distance (highest dot product) to the queries. In other words, the kNN returned are the (key, value) pairs that would have received the highest attention scores if we ran full attention over the memory.
> > >
> > > 2. Apologies for overlooking this point. We measure token-level perplexity, while Rae et al. measured the word-level perplexity.
> > >
> > > 3. Due to the lack of compute resources, we were not able to run the experiment to completion, but only to around 100k training steps so far. But we plan to add this ablation to the final version of the paper. We ran three models on the dataset arXiv-math, and here are the results (@100k steps):
> > >
> > > | Model | ArXiv-math |
> > > |---|---|
> > > | Transformer 2048 | 2.979 |
> > > | Transformer-XL 2048 | 2.625 |
> > > | Transformer-XL 512 + 1536 Memory | 2.858 |
> > >
> > > We can see that the vanilla transformer model with 2048 sequence length performed the worst in this comparison. We believe this is due to the fact that the tokens early in each subsequence can only see a limited window hence lacking of context information, whereas every tokens in the other two models always have the previous 2048 tokens as context. The Transformer-XL 2048 performed the best due to having the ability to backpropagate as well as attending for full layers.

---

### Official Review · Reviewer_gVad · 2021-11-02

**Correctness:** 4
**Technical Novelty And Significance:** 2
**Empirical Novelty And Significance:** 3
**Recommendation:** 6
**Confidence:** 4

**Main Review:**

Strengths:

- Integrating kNN module into Transformer is an interesting modeling technique, which would be potentially useful for language generation especially when the context is quite long.

- The kNN layer is integrated into the attention layer, while previous work usually integrates it into FFN blocks.

- The LM experiments are conducted on several corpora.

Weaknesses:

- The method is not compared with previous models in the experiments, such as Transformer-XL, Compressive Transformers, and other efficient Transformers for LM.

- There is no ablation study in terms of modeling.

- I didn't find clear comparisons of training speed and GPU memory usage between different models. It is valuable for the readers to have a sense of what kind of resource is needed to train or host such models.

- Is the application restricted to sequence decoding? Can we finetune the proposed model as BERT on downstream tasks, and how about the performance?

- The above question is related to generalization. Such kind of kNN method seems to suffer from too much memorization instead of generalization. If the model cannot generalize well, it would not be good at zero- or few-shot learning. I agree that language modeling is a meaningful task, but it is more like a testbed for modeling techniques that improve generalization. Otherwise, the shortcut of memorizing context would hide the real challenge.

- The code seems unavailable. It is unclear how many engineering efforts are required to reproduce the results. The paper writing ignores many details that are helpful for reproducing the method. Even the code would be released, the standalone paper should be informative enough.

- The kNN layer is densely triggered for each position, which seems quite heavy. How about using a gate to determine whether we need to perform kNN?

- Is there any evaluation about the performance of the learned kNN search? For example, we could conduct some human evaluations to see whether the retrieved knowledge makes sense for prediction.

- The model is only evaluated on the language modeling task. More tasks would be helpful to indicate whether the proposed method is general enough in terms of data distribution.


**Summary Of The Paper:**

The paper plugs a kNN memory module into a Transformer for long-distance knowledge. Specifically, the bottom layers are still Transformer blocks, then a kNN-augmented attention layer is used to store the processed (key, value) pairs into external memory, and another Transformer block is put at the top layer for aggregation. The LM experiments are conducted on webtext (C4), math papers (arXiv), books (PG-19), code (Github), as well as formal theorems (Isabelle).

**Summary Of The Review:**

I am okay to accept the submission. But it can be improved as mentioned in the above review (e.g., including ablation studies, more tasks, comparisons with previous models, and analysis).

---

> ### Author Response · Authors · 2021-11-18
> **Response to gVad**
>
> *“Is the application restricted to sequence decoding?”* and *"Can we finetune the proposed model as BERT on downstream tasks[...]?"* We agree that additional evaluation would further strengthen the experiments section, but we would also like to argue that language modeling already is a very diverse task. For example, the GPT papers have shown that language modeling alone equips models with a wide range of abilities. Liu et al. [1] have further shown that we can fine-tune decoder-only language models just as BERT for NLU tasks. Thus, we believe that general improvements to language modeling should carry over to other tasks as well. Instead of training the model on several different tasks, we demonstrate improvements across several datasets, which differ greatly in the kind of data they contain.
>
> *“The code seems unavailable. It is unclear how many engineering efforts are required to reproduce the results.”* We will open source the code as we stated in the reproducibility statement. The engineering efforts to implement our approach are moderate: the major changes to a standard Transformer training setup are (1) that we have to modify the data pipeline to feed the subsequences in the correct order (without shuffling), (2) that we augment the attention layer with the keys and values retrieved using an existing kNN algorithm (see Subsection 3.1), and (3) that we write the keys and values back to the memory after retrieval.
>
> *“I didn't find clear comparisons of training speed and GPU memory usage between different models.”* See our general answer on Speed and Memory Usage.
>
> *"The kNN layer is densely triggered for each position. How about using a gate to determine whether we need to perform kNN?"* We were discussing similar ideas, but decided to keep the approach as simple as possible for our initial publication.
>
> *“could conduct some human evaluations to see whether the retrieved knowledge makes sense for prediction”* We have manually investigated what the kNN model retrieved on arXiv, Github, and the Isabelle dataset. Please see Table 5,9,10 and the Appendix for some examples. We saw that the model achieves the cross-entropy gains by retrieving function/theorem names as well as their body.
>
> [1] Liu et. al., GPT Understands, Too.

---

> > ### Author Response · Authors · 2021-11-19
> > **Followup on our response.**
> >
> > Hi reviewer gVad, as the interactive discussion period is ending soon, we appreciate your time spent reviewing our manuscript and would be grateful if you can confirm whether our response addressed some of the concerns raised in the review? Please let us know if any issues remain and/or if there are any additional clarifications we can provide.

---

> > ### Comment · Reviewer_gVad · 2021-11-30
> > **thanks**
> >
> > Thanks for the reply. I acknowledge the response and update my detailed comments.

---

### Public Comment · ~Uri_Alon1 · 2021-11-09
**Questions**

Thank you for this paper!

The paper shows an elegant, simple and effective way to attend to far-away tokens in long documents, which is often necessary for practical settings.

I was left with three questions:

1. How can a memory of size 2k perform better (1.5 lower perplexity on Table 11!) than a standard transformer that was trained with a context of size 2k? Isn’t the context of size 2k strictly more expressive, and can learn anything that the 2k-memory learns, and even in a more precise way (because gradients do propagate to the 2k-away tokens, while in a memory they don’t)?
2. Times - the proposed method is impressive and surprisingly effective, but its straightforward limitation is time. How much time/throughput does training and testing take in tables 3+7+11? As one of the reviewers wrote, it is totally fine if it is slower, but it’s important to understand the tradeoff.
3. What would happen if you train the LM without memory, and incorporate the memory only at test time? Would it work somewhat better than the LM-without-memory? How far will it be from the LM that was trained *with* memory?

Thanks,

Uri

---

> ### Author Response · Authors · 2021-11-18
> **Thanks for your comments!**
>
> Regarding your questions:
>
> 1. Context size of 2k and memory of 2k are not strictly comparable, as the memory models have the memory of 2k tokens *in addition to* their context size of 512. As also suggested by the reviewers, we will make an apples-to-apples comparison, where we expect the memory to perform worse than the model with the longer context.
>
> 2. Execution time: See the general response.
>
> 3. Good question. Transformers often perform significantly worse, when we change the context length after training. We expect the same to happen when adding memory to a model without memory (the additional k=128 tokens effectively extend the length of the sequence we attend over). This is why we focused on a slightly different setup: train with a small memory and then test with a much larger memory.

---

### Author Response · Authors · 2021-11-18
**Common response**

We thank all reviewers for their thoughtful and constructive reviews. We feel encouraged by the positive comments on our work, e.g., **“the method is novel, intuitive and simple to implement”**, recognizing **the kNN module has its potential in long context modeling**, with **a comprehensive suite of datasets**. We also acknowledge that our submission can be improved in several ways and we will do our best to incorporate all suggestions made by the reviewers.

In this common response, we address the concerns shared by multiple reviewers, before we answer individual reviewer's questions in separate posts.

**Additional Baselines:** We fully agree that the paper would profit from additional experiments and baselines. We are running a new set of experiments where we compare the regular Transformer and the Transformer-XL both with and without memory. We already see that memory helps both models and across all datasets, but we are still waiting for all the runs to finish.

As suggested by reviewer 75XX, we are also using a larger model (12 layers instead of 6 layers), and have verified that the new perplexity results are comparable to other published results in the literature, and to the T5 model, which we use as baseline.

**Perplexities across datasets**

|Model| ArXiv-math | Github | Isabelle | C4 | PG-19 |
|---|---|---|---|---|---|
| Transformer-512 | 3.320 | 2.832 | 3.200 | 17.427 | 13.971 |
| Transformer-2048 | 2.798 | 2.485 | 2.440 | 15.394 | 12.503 |
| Transformer-512 XL | 2.756 | 2.255 | 2.511 | 15.379 | 12.554 |
| Transformer-512 + 8092 Memory | 2.591 | 2.133 | 2.278 | 15.287 | 12.782 |
| Transformer-512 XL + 8092 Memory | **2.452** | **1.966** | **2.198** | **14.629** | **12.158** |

**Hyperparameters:** Our choice of hyperparameters for the first version of our experiments relied mostly on prior experiments on natural language and formal mathematics. For all three types of models (vanilla transformer, transformer XL, and memory models) we have tried three different learning rates. If there are other hyperparameters that you would like to see explored in detail, we would be happy to follow your suggestions.

**Statistical significance:** We have started 3 runs for some of the experiments to report their standard deviation, which appears to be very low in our preliminary results (much lower than the gain of adding memory). The cost to run a statistically relevant number of runs for all experiments is prohibitive, unfortunately.  On math-arXiv at 120k training steps we observe the following variance:

Transformer XL:  2.904 ± 0.06

Transformer XL + 8k: 2.542 ± 0.05

**Speed and Memory Usage:** Memory overhead of kNN memory is relatively low, since it is not differentiable.  Computation overhead is higher; our current implementation of the memory on TPUs increases the step time by a factor of 2.4 (for Transformer-XL, 32k memory).
The runtime overhead on TPUs is dominated by the cost of the `gather` operation in kNN lookup, which is very slow on TPUs due to their memory architecture.  However, there are many different algorithms for kNN search, and our technique can work interchangeably with any of them.  Implementations of kNN search on CPUs (ScaNN library) or GPUs (Faiss library) would have very different overheads.  Both CPUs and GPUs are far more flexible than TPUs, and have much more efficient gather operations.  We do not believe that the computational cost of our current implementation, on TPUs, is representative of what could be achieved on other architectures.

**Perplexity measurements:** Unfortunately, we discovered a mistake in how we had computed perplexity. Our original numbers were computed from the cross-entropy loss with label smoothing turned on (with a coefficient of 0.1).  We have since been re-running all of the experiments with label-smoothing turned off to get the correct perplexity. While the exact numbers changed, memory still helps across all datasets. The numbers reported above used the corrected measurement.

[Edit log: merged the tables into one.]

---

> ### Author Response · Authors · 2021-11-23
> **Rebuttal revision and additional experimental results**
>
> We would like to thank all reviewers for their constructive feedback. We have uploaded a newer version of the paper, with changes including:
> - We added a paragraph describing the hardware used and speed and memory usage in Section 4.1.
> - We expanded on how we compute the datasets from publicly available sources in Sections 4.3 and 4.4, as well as the reproducibility statement.
> - We added references [CrossBatch], [REALM], and [Compressive Transformer].
>
> As we are still waiting for some experiments suggested by the reviewers, we decided not to update the experiments in the paper. However, we plan to add the following experimental results into the final version of the paper:
> - Adding the results given by the larger model (12 layers instead of 6 layers, as suggested by reviewer 75xx).
> - Adding more baseline comparisons to the Transformer-XL model, with various sequence lengths and memory sizes.
> - Adding an ablation study on standard deviation / statistical significance.
> - Adding an ablation study to compare memory and standard model while keeping the context window the same (512 + 1536 memory vs. 2048 Transformer-XL).
>
> Since the last update, we have obtained more results using the larger sized model and more comparisons to the baseline model. In particular, we now have numbers for Transformer-XL 2048, Transformer-XL 512 + 65k memory, and Transformer-XL 2048 + 8k memory.
>
> One can see that (i) 65k memory significantly improves upon 8k, and (ii) using memory helps even with longer sequence models (Transformer-XL 2048).
>
>
> |Model| ArXiv-math | Github | Isabelle | C4 | PG-19 |
> |---|---|---|---|---|---|
> | Transformer-512 | 3.320 | 2.832 | 3.200 | 17.427 | 13.971 |
> | Transformer-2048 | 2.798 | 2.485 | 2.440 | 15.394 | 12.503 |
> | Transformer-512 XL | 2.756 | 2.255 | 2.511 | 15.379 | 12.554 |
> | Transformer-512 + 8k Memory | 2.591 | 2.133 | 2.278 | 15.287 | 12.782 |
> | Transformer-512 XL + 8k Memory | 2.452 | 1.966 | 2.198 | 14.629 | 12.158 |
> | Transformer-512 XL + 65k Memory | **2.317** | **1.879** |  - | 14.08 |  **11.822** |
> | Transformer-2048 XL | 2.422 | 2.154 | - | 14.00 | 11.941 |
> | Transformer-2048 XL + 8k Memory | 2.321 | 1.914 | - | **13.76** | 11.834 |
>
> Lastly, we would like to update the standard deviation results. Also at 500k steps, the deviation between different runs of the same experiment appears to be much smaller than the gap between different models:
>
> Transformer XL: 2.746 ± 0.01
>
> Transformer XL + 8k: 2.445 ± 0.005

---

### Public Comment · ~Ayush_Agrawal2 · 2022-02-04
**Thanks for the interesting paper and Congratulations! It would be great if the authors could address the following questions**

1. Training vs. inference and the status of external memory
   We learn from section 3.1 and figure 1 that the external memory is populated by the (key, value) pairs generated during training.

   1.1 How is memory initialized? Is it empty at the beginning of training?

   1.2 What is the frequency of updates to the memory? Is it done at each iteration, or once after several iterations?

   1.3 "If the document is very long, old (key, value) pairs will be dropped from the memory to make room for new ones." What is the
          criterion to remove the key-value pairs from the external memory? Is it based on the time-stamps of the iterations or are there any
          other metrics used to discard them?

   1.4  Is the memory frozen after training or does it continue to be modified during inference? If it's frozen then how do you achieve the
          following goal mentioned in the introduction?: "For these tasks, the model must be able to work with large and continuously
          changing code repositories and knowledge bases, and it should be possible to utilize newly added code or facts immediately
          without the need for retraining or finetuning." If it's not frozen, then how is it modified during training?

2. kNN lookup: It is not clear how the distances for kNN are computed. Do you compute it just between key vectors or is it between the concatenation of key and value vectors? Is the distance L2 distance?

3. In the Appendix Section B, authors have mentioned the retrieval analysis. Can they explain what is the difference between the 'Retrieved Context' and 'Retrieved Token'.

---

> ### Public Comment · ~Yuhuai_Wu1 · 2022-02-04
> **Thanks for your interests! Let us know if you have further questions.**
>
> 1.1 Yes when we start read a new document, the memory is initialized as empty.
>
> 1.2 We update the memory once we finish reading the current chunk. So every iteration.
>
> 1.3 We implement a rolling buffer for the memory. So the earliest memory will be dropped. Improving the forgetting mechanism is definitely a very meaningful future research direction.
>
> 1.4 We clean the memory once we finish reading the document. During inference time, when we start reading a new document, we will again start from an empty memory, and keep adding new keys/values to it. This is the same for training as well as for inference.
>
> 2.1 We compute the attention scores between queries and keys using dot-product, and take the top-k.
>
> 3.1 Retrieved context just refers to the surrounding context (~10 tokens) of the retrieved token. Retrieved token is the token that actually got retrieved (as one of the top-k).
>
> P.S. We are currently preparing our camera ready version (also going to put it on arXiv soon). Stay tuned!

---

> > ### Public Comment · ~Ayush_Agrawal2 · 2022-02-05
> > **Thanks for the response!**
> >
> > Looking forward to the camera-ready version and code release (if possible)!

---

### Decision · Program_Chairs · 2022-01-20

**Decision:**

Accept (Spotlight)

**Comment:**

This paper studies the problem of dealing with long contexts within a Transformer architecture.
The key contribution is a  kNN memory module that works in concert with  a Transformer by integrating upper layers with additional retrieved context.

The idea is simple  but the execution is good.  While the  idea is reminiscent of other recent work on this topic, and novelty is somewhat borderline, it is practically useful.
Overall, though ambivalent, my recommendation is that the paper should probably be accepted